

# Historical gridded reconstruction of potential evapotranspiration for the UK

Maliko Tanguy[1], Christel Prudhomme[2,3,1], Katie Smith[1], Jamie Hannaford[1]

[1]NERC Centre for Ecology & Hydrology, Maclean Building, Benson Lane, Crowmarsh Gifford, Wallingford, Oxon, OX10 8BB, UK
[2]European Centre for Medium-Range Weather Forecasts, Shinfield Road, Reading, RG2 9AX, UK
[3]Department of Geography, Loughborough University, Loughborough, LE11 3TU, UK

*Correspondence to*: Maliko Tanguy (malngu@ceh.ac.uk)

**Abstract.** Potential Evapotranspiration (PET) is a necessary input data for most hydrological models and is usually needed at a daily or shorter time-step. An accurate estimation of PET requires many input climate variables which are in most cases not available prior to the 1960s for the UK, nor indeed most parts of the world. Therefore, when applying hydrological models for reconstructing earlier periods, modellers have to rely on PET estimation derived from simplified methods. Given that only monthly observed temperature data is readily available for the late 19th and early 20th century at a national scale for the UK, the objective of this work was to derive the best possible UK-wide gridded PET dataset with the limited data available.

To that end, firstly, a combination of (i) seven temperature-based PET equations, (ii) four different calibration approaches and (iii) seven input temperature data were evaluated against a gridded daily PET product based on the physically-based Penman-Monteith equation (the CHESS PET dataset), the rationale being that this provides a reliable 'ground-truth' PET dataset for evaluation purposes, given that no directly observed, distributed PET datasets exist. The performance of the models was also compared to the simplest possible estimation of PET ('naïve method') in the absence of any available climate data, the CHESS PET daily long term average (the period from 1961 to 1990 was chosen for this study), or CHESS-PET daily climatology. The analysis revealed that the type of calibration and the input temperature dataset had only a minor effect in the accuracy of the PET estimations at catchment scale. From the seven equations tested, only the calibrated version of the McGuinness-Bordne equation was able to outperform the 'naïve method' and was therefore used to derive the gridded, reconstructed dataset. The equation was calibrated using 43 catchments across Great Britain.

The dataset produced is a 5-km gridded PET dataset for the period 1891 to 2015, using as input data for the PET equation the Met Office 5-km monthly gridded temperature data available for that time period, which was disaggregated to daily temperature using pchip (piecewise cubic hermite interpolating polynomial) method. The dataset includes daily and monthly PET grids and is complemented with a suite of mapped performance metrics to help users assess the quality of the data spatially. The data can be accessed here: https://doi.org/10.5285/17b9c4f7-1c30-4b6f-b2fe-f7780159939c.



## 1 Introduction

Potential evapotranspiration is a conceptual variable which measures the meteorological control on evapotranspiration from an open surface water. Reference crop evapotranspiration (also referred to as Potential Evapotranspiration PET) is the rate of evapotranspiration of an idealised short grass actively growing and not short of water (Shuttleworth, 1993), providing an upper

limit of evaporative losses of grass. As evapotranspiration is a major factor in the catchment water balance (Beven, 2012), PET is used as input data to rainfall-runoff models.

Different formulations have been proposed for estimating PET. The most complex, combination methods, are based on physical processes accounting for the energy available to a plant to evaporate during photosynthesis, and the amount of water that can be dissipated in the atmosphere (Penman, 1948;Monteith, 1965). The simplest methods aim to capture the dominant

climatic factors in the plant evapotranspiration processes. The simplified methods can be broadly divided between the radiation-based methods (e.g. Doorenbos and Pruitt, 1984;Hargreaves and Samani, 1983;Jensen and Haise, 1963), which use measured data such as net solar radiation, sunshine hours or cloudiness factors; and the temperature-based methods (e.g. Blaney and Criddle, 1950;Thornthwaite, 1948;Oudin et al., 2005;McGuinness and Bordne, 1972), which use temperature as proxy for the radiative energy available, along with extra-terrestrial radiation estimated from the date of the year and latitude. Both

radiation and temperature methods are used instead of combination methods when the full set of climatic variables necessary for the latter is not readily available. However, there is no general agreement on the best performing method, and the final choice of the equation often depends on the application and data availability, as well as on the particular environmental setting (e.g. Donohue et al., 2010;Federer et al., 1996;Oudin et al., 2005;Prudhomme and Williamson, 2013;Xu and Singh, 2001;Xu and Singh, 2000). When little or no climatic variables are available, an alternative is to use remote sensing data in simplified

empirical PET methods (Barik, 2014;Barik et al., 2016;Knipper, 2017;Mu, 2013), however this can only be applied to the satellite era (from the 1970s-1980s).

In the UK, the Met Office Rainfall and Evaporation Calculation System (MORECS Thompson et al., 1982) is one of the main sources of PET estimates, available as an approximately 40 x 40 km monthly gridded product, with time series from 1961.

MORECS is based on the Penman-Monteith formulation (Monteith, 1965), but includes evaporation from rainfall intercepted by the canopy and considers 14 different vegetation types and three different soils (Hough, 2003). Recently, the Centre for Ecology and Hydrology published the Climate, Hydrology and Ecology research Support System (CHESS), a 1km gridded daily meteorological and land state dataset for Great Britain (Robinson et al., 2016a; 2016b; 2017) spanning the period 1961-2015. It includes PET data calculated from the meteorological variables using the Penman-Monteith equation for a well-

watered grass surface, following the Food and Agriculture Organisation of the United Nations (FAO) guidelines for computing reference crop PET (Allen, 1998), both with and without corrections for water intercepted by the canopy (Robinson et al., 2016b).



In the UK, PET data is mainly used for hydrological modelling, where streamflow time series are generated from rainfall and PET inputs. This is particularly useful in providing information where streamflow observations do not exist, i.e. in reconstructing flows for pre-observational periods, or to explore the response of a changing climate on hydrology. Currently, there is no readily available source of PET time series for studying long-term variability and change in hydrological regimes

before the 1960s, including water resources availability and drought patterns. This is a major obstacle, because historical drought periods are used in water resources and drought planning (e.g. Watts et al., 2012), as well as for providing a baseline of past hydrological variability for future change assessments. In practice, however, limited availability of atmospheric variables makes it difficult to account for the majority of evapotranspiration processes for the pre-1960 period using the Penman-Monteith equation. Simpler methods therefore need to be used as alternative, but this requires a thorough evaluation

of the differences they bring when compared with established datasets. This study focuses on temperature-based PET equations as temperature (together with precipitation) is the climate variable that has observed, spatially distributed records for the longest period for the UK. While the focus was on applying methods for historical reconstruction, this undertaking will provide useful information for other applications where only temperature are available; for example, in hydrological forecasting or some long-term climate change impact studies.

This paper describes the derivation of a 5km gridded daily and monthly PET dataset for the United Kingdom from 1891 to 2015, for use in hydrological modelling. First, the data used for calibration, validation and production of the gridded dataset are presented. This is followed by the methods, where the temperature-based PET equations, the calibration strategies and the evaluation approach used are described. Thirdly, the results of the evaluation of the PET equations, and the assessment of the

final PET grids are presented, and lastly, limitations of the product and recommendations for users are listed.

## 2 Data

### 2.1 Temperature data

Three main sources of high resolution gridded national-scale temperature data exist for the UK:

- CHESS-met high resolution mean daily temperature (1-km grids, daily time series) for 1961-2015 for Great Britain

(CHESS-temp daily). This dataset is part of a larger dataset developed by CEH for environment modelling applications; its derivation is fully described in Robinson et al. (2016a).

- UKCP09 mean monthly temperature (5-km grids, monthly time series) for 1910-2015 for the UK, including Northern Ireland (UKCP09-temp monthly). This is part of a larger dataset developed by UK Met Office, its derivation is fully described in Perry and Hollis (2005).

- HistDrought mean monthly temperature (5-km grids, monthly time series) for 1891-1909 for the UK, including Northern Ireland (HD-temp monthly). This was derived using the same methodology as UKCP09, using historic



weather station data rescued by the Met Office in the Historic Droughts project (NERC grant number: NE/L01016X/1).

The two latter were combined in this study and treated as a single dataset to provide a single, continuous record of temperature from 1891-2015, which is used to derive the long-term Potential Evapotranspiration dataset that is the focus of this study. The combined dataset will be referred to as UKCP09 in the rest of the paper. The shorter (1961-) CHESS dataset is used for calibration and sensitivity testing.

Because of the coarser temporal and spatial resolution of temperature data prior to 1961, alternative datasets were generated and used in the analysis to quantify the sensitivity of PET derivation to temperature input:

- ▪ CHESS daily mean temperature climatology (1-km grids) (CHESS-temp clim): long term average (1961-1990) of daily mean temperature, derived from CHESS-temp daily.  This provides a default option that could be used even if no temperature data were available in the past (or future).
- ▪ CHESS daily mean temperature derived from monthly averages (1km grids):
    - (i) Constant during the month (CHESS-temp monthly I). This means there are step changes in temperature between consecutive months.
    - (ii) Interpolated using pchip (piecewise cubic hermite interpolating polynomial) method for a smooth transition between months (CHESS-temp monthly II). Pchip was selected because (i) the fitted curve passes through observed values at inflexion points unlike spline or quadratic methods, for example, and (ii) it does not require re-fitting when the period of application is extended.
    - (iii) Disaggregated to daily using CHESS daily mean temperature climatology pattern (CHESS-temp monthly III). The daily relative variation in temperature follows the climatology, but the values are corrected so that monthly mean temperatures are correct.
- ▪ UKCP09 daily mean temperature derived from monthly averages:
    - (i)     Constant during the month (UKCP09-temp monthly I).
    - (ii)    Interpolated using pchip method (UKCP09-temp monthly II).

In summary, seven different daily temperature datasets were used as input data to the temperature-based PET equations: CHESS-temp daily, CHESS-temp clim, CHESS-temp monthly I, CHESS-temp monthly II, CHESS-temp monthly III, UKCP09-temp monthly I and UKCP09-temp monthly II.

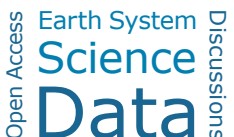

## 2.2 PET

One main source of national-scale mean daily PET time series was available: CHESS-PET 1-km grids, daily time series (Robinson et al., 2016b) available for 1961-2015, calculated using the Penman-Monteith (PM) equation (Monteith, 1965) for FAO-defined well-watered grass (Allen, 1998). Because the PM equation is a physically-based model which combines the

energy balance with the mass transfer method, and is recommended by the FAO to calculate PET, CHESS-PET daily is here considered as "ground truth" and proxy for observations (referred to as CHESS-PM hereafter).

CHESS-PM daily climatology (CHESS-PM climatology) was also calculated from 1961 to 1990, and is used as a 'naïve method' against which the PET reconstruction methodology can be tested to assess performance.

## 3 Methods

To produce the PET gridded reconstruction product, first a set of seven temperature-based PET equations were evaluated. These were tested using four different calibration strategies (in addition to the non-calibrated equations), and seven different temperature input datasets (section 2.1). Once the best combination of equation/calibration strategy/temperature input data was selected, the actual gridded PET reconstruction was produced. Quality assessment tests were then performed on the final gridded product, using a range of performance metrics to quantify the reliability of the product in different locations. Figure 1

summarises the different steps of the work.

### 3.1 Temperature-based equations

This work aims to generate gridded PET daily time series for the UK back to the last decade of the 19[th] century to use as input to hydrological models for reconstructing streamflow time series. Because temperature data is the only climate data available continuously at the UK national scale over this time frame, the work has focused on testing and developing temperature-based

equations for application on the UK. Four main temperature-based equations were evaluated (see Table 1, Eq 1-4): Hamon (Hamon, 1961), McGuinness-Bordne (McGuinness and Bordne, 1972), Blaney-Criddle (Blaney and Criddle, 1950) and Kharrufa (Kharrufa, 1985). Each contains a number of parameters representative of the climatic region where the equation was originally developed, which can be modified through a calibration procedure to match the climatic regime of the UK. Three additional equations were also assessed, that were not suitable for simple calibration techniques, or had set calibrations: Oudin

(Oudin et al., 2005), MOHYSE (Fortin, 2006) and Thornthwaite (Thornthwaite, 1948) (Eq5 to 7 in table 1).

Note that equations requiring minimum and maximum temperature (Droogers and Allen, 2002;Hargreaves, 1983;Heydari and Heydari, 2014) were not considered here as they are not always available outside the observation period (e.g. from forecasts).



### 3.2 Calibration strategies

To compare the different calibration methods in a time-efficient way, the calibration and testing was done at catchment scale using two independent sets of catchments representative of typical hydroclimatic conditions prevailing in the UK and with good spatial coverage: 43 were used for calibration and assessment of the equations, and an additional 263 (making a total of

306 catchments) used for evaluation of the final PET grids (Figure 2). The spatial averaging of temperature and PET time series to conduct the analysis has the advantage of smoothing out any discontinuity that could exist at the grid-scale level due to different interpolation algorithms and recording stations and which could impact on local performance of the PET generation technique. In addition, for many practical hydrological modelling applications, PET is required at the catchment scale.  The impact of catchment-scale vs grid-scale calibration is assessed in section 4.2.

As previously mentioned, temperature-based PET equations use parameters to link temperature to PET as a simplification of full evaporation dynamics. Because of important climatic variation across space, and in time across the year, it might be possible that optimal parameterisation could be achieved by letting the parameters vary in time and space. Therefore, four calibration strategies were considered, which are graphically represented in Fig. 3: from a global parameterisation leading to

a single equation for all 43 catchments, to a local and monthly parametrisation leading to 12 equations for each of the 43 catchments. The trade-off between a simplified method (global parameterisation), which is much easier to implement, and a local method, that requires a long calibration procedure and a parameter transfer methodology for Northern Ireland (where no daily PET dataset is available), is discussed in the results section.

Two independent time periods of similar length were selected for the calibration (1961-1990) and evaluation (1991-2012) procedures. The equations' parameters were calibrated using the ordinary least squares method against CHESS-PM.

### 3.3 Evaluation

#### 3.3.1 Catchment-scale performance metrics for evaluating the combinations of PET equations/calibration
**strategy/input temperature data**

This evaluation corresponds to the upper part of the diagram in Fig. 1, and was done on the 43 calibration catchments shown in Fig.2a for the period 1991-2012.

Two metrics were used to evaluate the best combination of forcing data, PET equation, and calibration strategy: the Mean Absolute Percentage Error (MAPE) and Nash-Sutcliffe Efficiency (NSE) coefficient, using CHESS-PM daily as ground-truth.


MAPE is widely used in the forecasting community to evaluate accuracy of output from models (Danladi et al., 2017;Lefebvre and Bensalma, 2015). Applied to PET, it is calculated as:



$$MAPE = \frac{100}{n} \sum_{t=1}^{n} \left| \frac{PET_O^t - PET_m^t}{PET_O^t} \right|$$

where $t$ is the time-step, $n$ is the number of time-steps, $PET_O^t$ is the actual value of PET at time $t$ and $PET_m^t$ is the modelled value of PET at time $t$.

In order to be able to apply MAPE, values of observed PET of 0 were replaced by 0.1. Smaller values of MAPE indicate greater accuracy of the model prediction.

NSE was developed to assess hydrological models (Nash and Sutcliffe, 1970), but the concept was here applied to assess the PET models. It is calculated as:

$$NSE = 1 - \frac{\sum_{t=1}^{n}(PET_m^t - PET_O^t)^2}{\sum_{t=1}^{n}(PET_O^t - \overline{PET}_O)^2}$$

where $\overline{PET}_O$ is the mean of observed PET, $PET_m^t$ is modelled PET at time t, and $PET_O^t$ is observed PET at time t.

Nash–Sutcliffe efficiency can range from $-\infty$ to 1. An efficiency of 1 (NSE = 1) corresponds to a perfect match of modelled PET to the observed data. An efficiency of 0 (NSE = 0) indicates that the model predictions are as accurate as the mean of the observed data, whereas an efficiency less than zero (NSE < 0) occurs when the observed mean is a better predictor than the model or, in other words, when the residual variance (described by the numerator in the expression above), is larger than the data variance (described by the denominator).

The performance of each of the different combinations (PET equations/calibration approaches/input temperature data) was compared against an independent benchmark (reference) for comparison - CHESS-PM clim, used as an alternative way to estimate daily PET locally when no data is available (e.g. for the past or the future). It is worth noting that $\overline{PET}_O$ used in the calculation of NSE is different from CHESS-PM clim, in that (i) the latter has a daily value for each day of year (which is repeated for every year), whereas $\overline{PET}_O$ is just a single value (CHESS-PM averaged over time); and (ii) CHESS-PM clim is the daily average PET calculated for the period 1961-1990, whereas $\overline{PET}_O$ is the average CHESS-PM value for the evaluation period which is 1991-2012.

### 3.3.2 National-scale performance metrics for final PET grids quality assessment

To assess the quality of the final gridded PET product, a series of performance metrics were calculated at national scale, and provided together with the final product. These are the metrics which were used in the steps shown in the second part (bottom half) of Fig. 1. Once again, CHESS-PM daily was used as ground-truth.

In addition to MAPE and NSE described in the previous section, the following four metrics were also calculated:

- Correlation coefficient ($r$)





- Bias ratio ($\beta$), calculated as: $\beta = \frac{\overline{PET_m}}{\overline{PET_O}}$ where $\overline{PET_m}$ and $\overline{PET_O}$ are the mean modelled and observed PET.

- Variability ratio ($VR$), calculated as: $VR = \frac{CV_m}{CV_O} = \frac{\sigma_m / \overline{PET_m}}{\sigma_O / \overline{PET_O}}$ where CV is the coefficient of variation and σ is the standard deviation of PET.

- Kling-Gupta efficiency (KGE) which is a combination of $r$, $\beta$ and $VR$ (Gupta et al., 2009;Kling et al., 2012) and is calculated as: $KGE = 1 - \sqrt{(r-1)^2 + (\beta-1)^2 + (VR-1)^2}$

KGE, VR, $r$ and $\beta$ coefficient all have their optimum at unity (1).

## 4 Results

### 4.1 Assessment of the temperature-based PET equations

In this section, results from the evaluation represented in the first part (upper half) of Fig. 1 are presented.

Figure 4 shows a summary graphic showing the average MAPE and NSE for all combinations of forcing data, PET equation and calibration strategy tested. For simplicity, Figure 4 does not show the results from:

- uncalibrated models (as these were performing worse than the calibrated models);

- using CHESS-temp monthly III forcing (similar results to those for CHESS-temp monthly II);

- using UKCP09-temp monthly I and II, as they were only used with the final selected equation as an additional test to check the effect of spatial resolution on the results.

The full list of performance metrics is given in tables A1 and A2 in the supplementary information.

Figure 4 shows that:

(i)     Calibration yields substantial improvement in performance, except for Hamon (Eq1 in table 1) which performed well before calibration.

(ii)    Calibration strategy has very little effect on the performance. Both annual and global calibrations show a similar performance to the locally-calibrated, monthly models. The simplest calibration approach was hence adopted: national-scale application was conducted using the 1P-GB strategy.

(iii)   Daily temperature forcing only performs marginally better than forcing based on monthly temperature time series. This might be explained by the small day-to-day variability in temperature fields (and hence, in any resulting PET field) compared with other climate variables such as wind speed, humidity or radiation, which provide a much larger contribution to the daily variability of PET than temperature. The artificial daily pattern introduced by temporal disaggregation of monthly temperature is in fact small compared with the error introduced by using temperature-only forcing to estimate PET. This is illustrated in Figure A1 (supplementary material).



(iv)     CHESS-PM climatology is only outperformed by the calibrated version of McGuinness-Bordne equation (Eq2 in table 1). This suggests a small inter-annual variability of PET, with a daily climatology being a good alternative when no other time series available. Note however that the evaluation period (1991-2012) is too short for investigating the possible impact of trends (e.g. temperature trends, interdecadal variability, climate change signal)

in the PET signal, which might reduce the overall ability of a climatology average to represent PET correctly. Also worth noting is that, surprisingly, the results also suggest that, in the absence of any climate data available, it would actually be more advisable to use CHESS-temp clim (long-term daily temperature climatology) and run the data through the calibrated McGuinness-Bordne equation rather than using CHESS-PM climatology. NSE scores are equivalent for both approaches but MAPE is worse for the latter. This is because CHESS-PM climatology produces

a much smoother time series than running McGuinness-Bordne equation using CHESS-temp clim, which amplifies the daily fluctuations seen in the temperature climatology, and produces a PET estimate closer to the observed.

In conclusion, a single annual McGuinness-Bordne PET equation calibrated over all catchments simultaneously was selected as the most appropriate method to derive daily PET time series from monthly mean temperature data.

To investigate the effect of coarser spatial resolution in the forcing temperature data, McGuinness-Bordne 1P-GB was applied using UKCP09-temp monthly I and II (5km gridded data) as forcing data. Results show (table A1 in supplementary material) that at the catchment scale, the spatial resolution of the forcing temperature data has virtually no effect on the performance, with MAPE and NSE values almost identical when using 1-km gridded CHESS-temp monthly I data (MAPE=32.02,

NSE=0.72) or 5-km gridded UKCP09-temp monthly I data (MAPE=31.65, NSE=0.72), and when using CHESS-temp monthly II data (MAPE=32.13, NSE=0.72) or UKCP09-temp monthly II data (MAPE=32.06, NSE=0.72). This suggests that for the reconstruction prior to 1961, when only 5km monthly temperature time series are available across the UK, performance is expected to be equivalent than if finer spatial and temporal mean temperature resolution data existed.

### 4.2 Assessment of the final PET grids

This section presents the results of the assessment described in the steps shown in the second part (bottom half) of Fig. 1. Based on the results in section 4.1, the McGuiness Bordne 1P-GB equation calibrated on 43 catchments was selected to generate a 5-km PET dataset covering the period 1891 to 2015, using UKCP09-temp monthly II data. A monthly version of the dataset (monthly aggregation of the daily PET for consistency) was also produced for applications requiring a coarse temporal resolution such as groundwater modelling, which has the advantage of a smaller data volume.

One of the possible issues with a catchment-scale calibration such as implemented here is its applicability at a finer spatial scale. Three tests, which are summarised in Fig. 1 (bottom half), were conducted to assess the gridded product:



- TEST 1: catchment-average daily PET time series extracted from the final 5-km daily PET gridded product were compared with daily PET series based on catchment average temperature, derived using the same equation to test the validity of a catchment-scale calibration.

- TEST 2: catchment-average daily PET extracted from the final 5-km daily PET gridded product compared with
catchment-averaged CHESS-PM (ground-truth), both at daily and monthly timescale, to evaluate the performance of the final reconstructed product on a catchment-scale basis.

- TEST 3: grid-scale final 5-km daily PET gridded product compared with CHESS-PM regridded at 5-km, both at daily and monthly timescale, to evaluate the performance of the final reconstructed product on a grid-scale basis, as opposed to the catchment-scale tests above.

For TEST 3, each metric was calculated for each single grid cell and for MAPE for each month as the error varies seasonally. Monthly MAPE can be used as a measure of the uncertainty in the data.

The evaluation was conducted over the 1991-2012 period using the metrics described in section 3.3., and shown in Fig. 1.

At catchment-scale (TEST 1) there is virtually no difference in deriving PET time series from a gridded PET or from PET calculated with the same equation using catchment-average temperature, with a correlation coefficient exceeding 0.999962 for the 306 catchments. This validates our assumption that the selected equation calibrated at catchment scale is applicable at grid-scale.

When comparing the new PET dataset with CHESS-PM at catchment scale (TEST 2), the results are more varied. Spatial differences can be observed between the different metrics and are represented in detail in figures A2 and A3 of the supplementary material. The results are not discussed here as they are very similar to the grid-to-grid comparison (TEST 3) described next.

At grid-scale (TEST 3), the following observations can be made (Fig.5 for daily PET and Fig. 6 for monthly PET; note differences in the legend colour scale):

(i)     Performance is greater for monthly (Fig.6) than daily (Fig.5) PET, except for Bias ratio β which is very similar for both. This suggests that the error is greater at daily than at monthly resolution, likely to be due to the smoothing of the day-to-day variability absent in a temperature-based method. The PET equation shows very good performance

at monthly scale with values of NSE > 0.9, $r$ > 0.97 and KGE > 0.8 for the whole country. For daily values, the performance is more moderate.

(ii)     Performance varies spatially, but this variability depends on the metrics chosen and is different for monthly and daily PET. For daily PET, MAPE (Fig.5a), NSE (Fig.5b) and $r$ (Fig.5c) show lower performance near the coasts. This is probably because daily variation of wind and humidity are higher near the coast, which is not captured in



temperature-based PET equations and hence resulting in larger errors. VR (Fig.5d) displays a North-South gradient in performance, the North being better. This is because the coefficient of variability in observed PET is smaller in the North than it is in the South, with less daily noise (see figure A1 in supplementary material, grey line). The Bias ratio β (Fig.5e) is close to 1 everywhere across Great Britain, which indicates that the calibrated McGuinness-Bordne equation shows very little bias. KGE (Fig.5f) which is a combination of r, VR and β, shows a North-South gradient as the strongest influence comes from VR. For monthly PET, the daily noise in the climate variables is absent, which explains smaller differences in performance scores for most metrics.

The metrics grids are provided as part of the dataset, and can inform the users on the quality of the PET estimation for a given location.

## 5 Limitations and recommendations

The main limitation of this PET dataset comes from the method used to derive it, which only takes temperature into account. This is the case particularly for the daily PET dataset presented here. The PM evapotranspiration equation has radiative and convective components. In simplified temperature-based equations, temperature is used as a proxy for radiation but does not account for the convective aspect. Therefore, temperature-based equations are not able to reproduce the full daily fluctuation of PET, and are only a smoothed version of reality. Numerous previous studies (e.g. Seiller and Anctil, 2016) show that the choice of PET equation can have an important impact on hydrological outputs.

Datasets based on physically-based equations such as CHESS-PM would most likely be a better option when- and where they are available, which is not the case in Great Britain before 1961, and in Northern Ireland. When such high-resolution physically-based PET datasets are not available, temperature-based PET datasets such as the historical PET dataset reconstructed here provide a valuable substitute. As suggested by Oudin et al. (2005), such temperature-based methods are suitable for conceptual hydrological modelling, and when available at a fine spatial scale, also suitable for distributed hydrological modelling. The gridded McGuinness-Bordne PET dataset derived here is unique for the UK, with a high spatial (5km) and temporal (daily) resolution covering 125 years over the UK, including Northern Ireland. By capturing monthly trend, McGuinness-Bordne PET is suitable for deriving monthly river discharges or run-offs, and its daily temporal resolution is sufficient for most hydrological modelling applications. However, caution is recommended when calculating the daily water balance as the inability to account for daily fluctuation in PET can yield inaccurate results at daily scale. Users are strongly advised to look at performance metrics associated with the dataset in their study area, such as for example monthly MAPE, which provide information on the uncertainty in the estimates. Note that because of the absence of the reference dataset CHESS-PM in Northern Ireland, no quality metrics are available in that region.



Beyond generating a new 125years gridded daily PET dataset for the UK, this research has highlighted valuable insights for PET calculation in the UK: (i) calibration is essential for realistic results, but the choice of calibration method (global/annual or local/monthly) has a minimal effect, therefore the easiest, more cost-effective calibration method is recommended (global/annual); (ii) the temporal resolution of the input temperature data and the temporal disaggregation method when using

monthly data has little influence on the results; (iii) the temperature-based PET equation (from the seven equations tested) that produces the best results for the UK is the calibrated version of the McGuinness-Bordne equation; (iv) for this equation, the spatial resolution (1km or 5km) of the input temperature data has virtually no effect in the results at catchment scale; (v) CHESS-PM daily climatology is the second best of the tested options, and therefore is a possible alternative source of PET if no climate variables are available. Whilst mean seasonal PET or climatology can been used in hydrological modelling

(Burnash, 1995;Calder et al., 1983;Fowler, 2002), McGuinness-Bordne derived PET time series are preferable as they are able to reproduce the inter-annual variability existing in PET, absent from any climatology; (vi) temperature-based equations perform better at monthly scale than at daily scale, as the full daily fluctuation of PET due to other climate variables (wind speed, humidity, radiation) are not being accounted for, but these are smoothed out at monthly scale; and finally, (vii) performance of the McGuinness-Bordne equation across the UK is variable in space, and the metrics grids provided within the

dataset can inform future work on the adequacy of using this approach for estimating PET in particular areas.

**6 Data access**

The new PET dataset is called "Historic Gridded Potential Evapotranspiration (PET) based on temperature-based equation McGuinness-Bordne calibrated for the UK (1891-2015)" and is available from https://doi.org/10.5285/17b9c4f7-1c30-4b6f-b2fe-f7780159939c. The dataset is stored in NetCDF4 format.

For the monthly grids, the dataset is structured as three-dimensional grids covering the United Kingdom, with twelve time-steps (monthly grids) in the time dimension in each yearly file, and a spatial resolution of 5-km.

For the daily grids, the dataset is structured as three-dimensional grids covering the United Kingdom, with 365 or 366 (leap year) time-steps (daily grids) in each yearly file, and a spatial resolution of 5-km.

In addition, four metric files, also in NetCDF format, accompany the PET files (two for daily grids and two for monthly grids),

also at a spatial resolution of 5-km.

The data are projected using the British National Grid co-ordinate system.

The following citation should be used for every use of the data: Tanguy, M.; Prudhomme, C.; Smith, K.; Hannaford, J. (2017). Historic Gridded Potential Evapotranspiration (PET) based on temperature-based equation McGuinness-Bordne calibrated for

the UK (1891-2015). NERC Environmental Information Data Centre.

The dataset is available for download from the CEH Environmental Information Data Centre (EIDC).




## 7 Acknowledgement and data source

This is an outcome of the IMPETUS (grant number: NE/L010267/1) and Historic Droughts (grant number: NE/L01016X/1) projects, funded by the Natural Environment Research Council.

The authors would like to thank their CEH colleague Cath Sefton for offering valuable feedback on the manuscript, and the

Met Office, in particular Mark McCarthy and Tim Legg for providing the historic monthly temperature data.

CHESS temperature data (Robinson et al., 2016a) can be downloaded from the EIDC catalogue: https://catalogue.ceh.ac.uk/documents/b745e7b1-626c-4ccc-ac27-56582e77b900

CHESS PET data (Robinson et al., 2016b) can be downloaded from the EIDC catalogue: https://catalogue.ceh.ac.uk/documents/8baf805d-39ce-4dac-b224-c926ada353b7

UKCP09 temperature data (Perry and Hollis, 2005) can be downloaded from CEDA catalogue: http://catalogue.ceda.ac.uk/uuid/87f43af9d02e42f483351d79b3d6162a

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

25

30

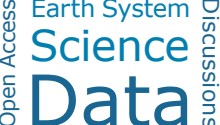

**Table 1: Temperature-based equations evaluated in this study**

| Equation number and name | Formula | Abbreviations | Reference | Comments | Calibrated in this study |
|---|---|---|---|---|---|
| Eq1: Hamon | Version 1:<br>$PE[inches\ day^{-1}] = 0.55\left(\frac{N}{12}\right)^2 \rho_s$<br>Version 2:<br>$PE[mm\ day^{-1}] = \left(\frac{N}{12}\right)^2 exp\left(\frac{T}{16}\right)$ | ▪ $N$ maximum possible daylight hours [h]<br>▪ $\rho_s$ saturated vapour density [g m⁻³]<br>▪ $T$ average temperature [ºC] | Version1:<br>Hamon (1961)<br><br>Version 2:<br>Oudin et al. (2005) | Originally developed for the USA.<br>Version 2 was used in this study. | YES |
| Eq2: McGuinness-Bordne | $PE[mm\ day^{-1}] = \frac{1}{\lambda}S_0\left(\frac{T+5}{68}\right)$ | ▪ $\lambda$ latent heat of vaporisation [MJ kg⁻¹]<br>▪ $T$ temperature [ºC]<br>▪ $S_0$ extraterrestrial radiation [MJ m⁻² day⁻¹] | McGuinness and Bordne (1972)<br>Oudin et al. (2005) | Originally developed for the USA.<br>Oudin et al. (2005) version used in this study | |
| Eq3: Blaney-Criddle | $PE[mm\ day^{-1}] = kTp_d$<br>with $p_d = 100\frac{N_d}{\sum_{i=1}^{365} N_i}$ | ▪ $p_d$ mean daily percent of annual daytime hours for day $d$<br>▪ $T$ mean air temperature<br>▪ $k$ monthly consumptive use coefficient.<br>▪ The coefficients $k$ depends on crop, location and season | Blaney and Criddle (1950) | Originally developed to estimate the irrigation requirements of crops in Western USA. | |
| Eq4: Kharrufa | $PET[mm\ day^{-1}] = 0.34\left(100\cdot\frac{DL}{365\times12}\right)\cdot T_a^{1.3}$ | ▪ DL day length [h day⁻¹]<br>▪ $T_a$ air temperature [ºC] | Kharrufa (1985) | Originally developed for arid regions. | |
| Eq5: Oudin | $\begin{cases} PE[mm\ day^{-1}] = \frac{1}{\lambda}S_0\left(\frac{T+5}{100}\right) & if\ T > -5℃ \\ PE[mm\ day^{-1}] = 0 & if\ T \leq -5℃ \end{cases}$ | ▪ $\lambda$ latent heat of vaporisation [MJ kg⁻¹]<br>▪ $T$ temperature [ºC]<br>▪ $S_0$ extraterrestrial radiation [MJ m⁻² day⁻¹] | Oudin et al. (2005) | Version of McGuinness-Bordne equation calibrated for catchments in Australia, USA and France, and which was assessed as the best temperature-based PE equation following a review of various PE methods for use as input to hydrological models (Oudin et al., 2005). | NO |
| Eq6: MOHYSE | $PET = \frac{4.088}{\pi}\cdot\omega\cdot exp\left(\frac{17.3T_a}{238+T_a}\right)$ | ▪ ω the sunset hour angle [rad]<br>▪ $T_a$ air temperature [ºC] | Fortin (2006) | Developed in Quebec. | |
| Eq7: Thornthwaite | $PE'[mm\ month^{-1}] = 16\left(\frac{DL}{360}\right)\left(\frac{10T_a}{I}\right)^K$<br>$a = 0.49239 + 0.01792\ I - 7.71\ 10^{-5}I^2$<br>with $I = \sum_m^{12}\left(\frac{T_m}{5}\right)^{1.514}$ annual heat index | ▪ $T_a$ air temperature [ºC]<br>▪ $T_m$ mean temperature of month $m$ [ºC] | Thornthwaite (1948) | Thornthwaite correlated mean monthly air temperature with PE as determined by water balance studies in valleys of east-central USA. | |





**Figure 1: Schematic diagram of the process for identifying the optimal method to derive a UK-scale gridded daily PET product.**



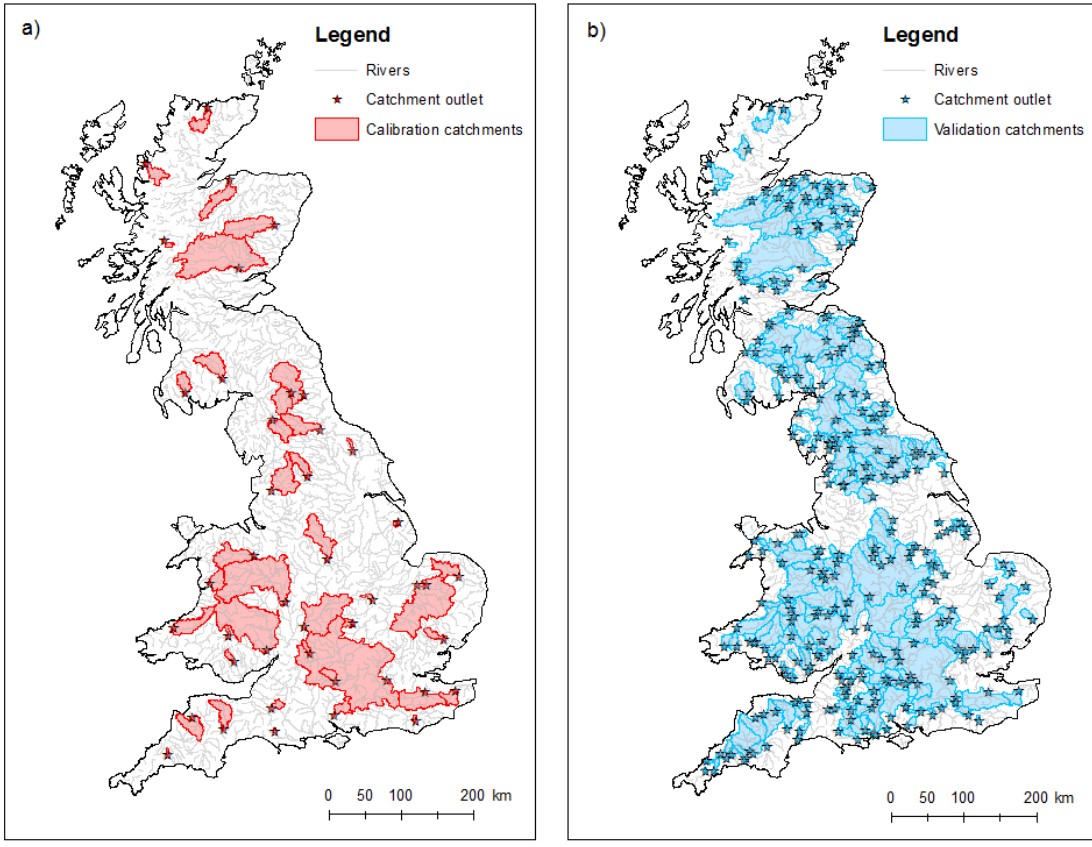

**Figure 2: Location of the catchments selected for a) calibration and assessment of the PET equations; b) evaluation of the final PET grids**



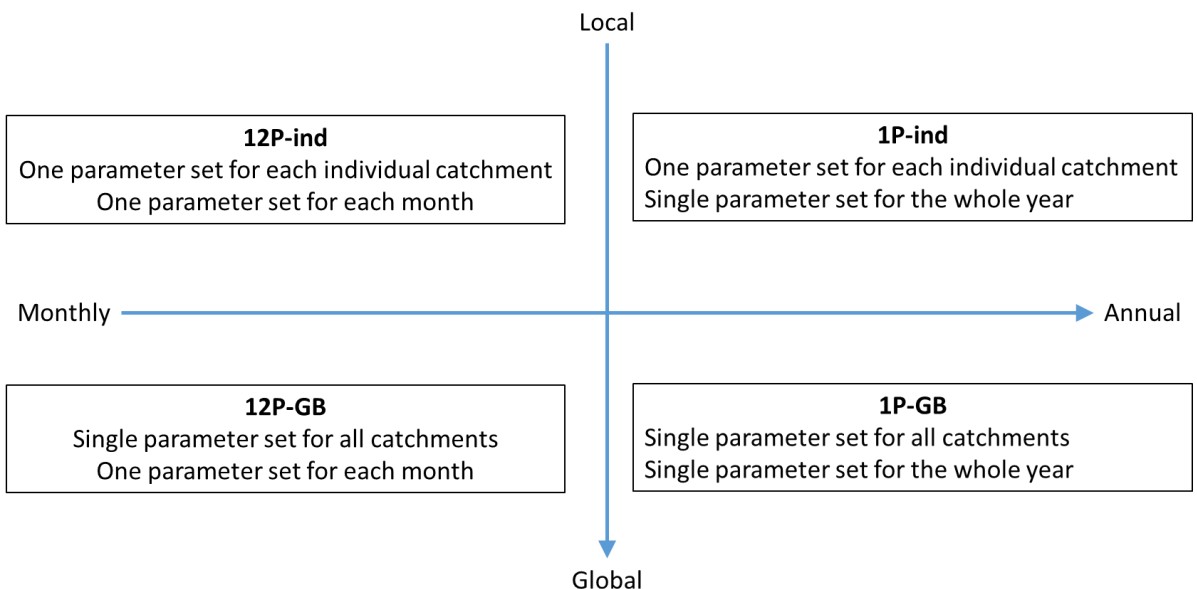

**Figure 3: Schematic of the calibration strategies**

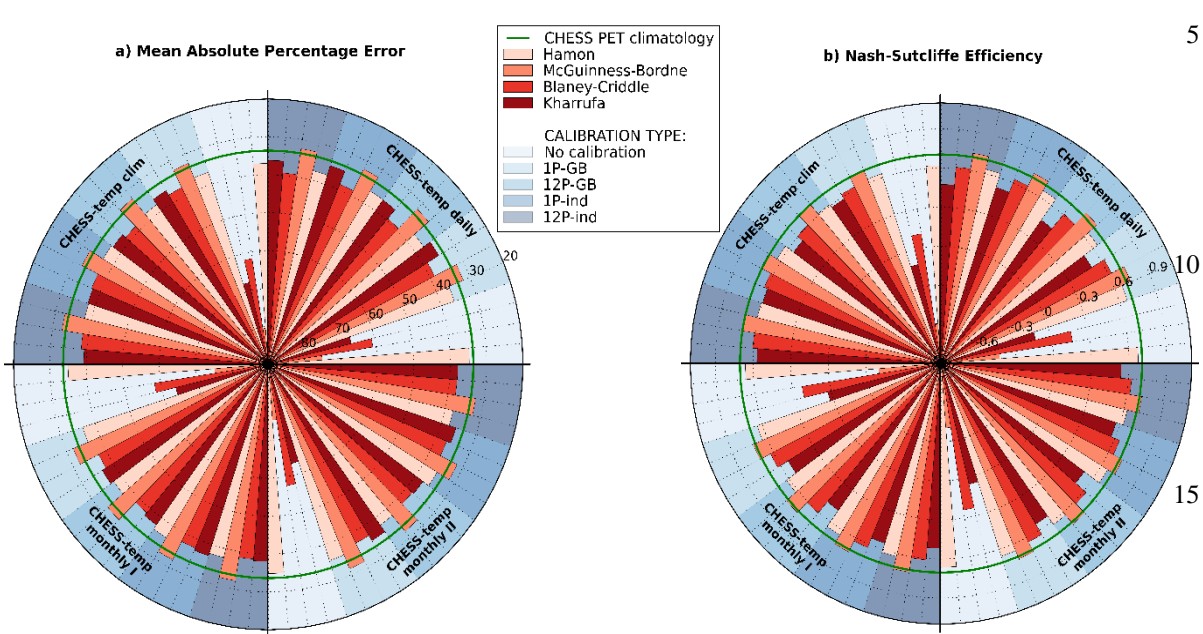

**Figure 4: Performance of the different combinations of PET equations (shown in different shade of red), calibration approaches (shown in different shade of blue) and input temperature data (one in each quadrant). The green line on the plots shows the reference CHESS PET climatology for comparison (a) Mean Absolute Percentage error (MAPE) – note that the y axis is inverted so that lower MAPE values (which indicates better performance) are shown towards the outside of the radial plot. (b) Nash-Sutcliffe Efficiency (NSE). Higher NSE indicates better performance.**





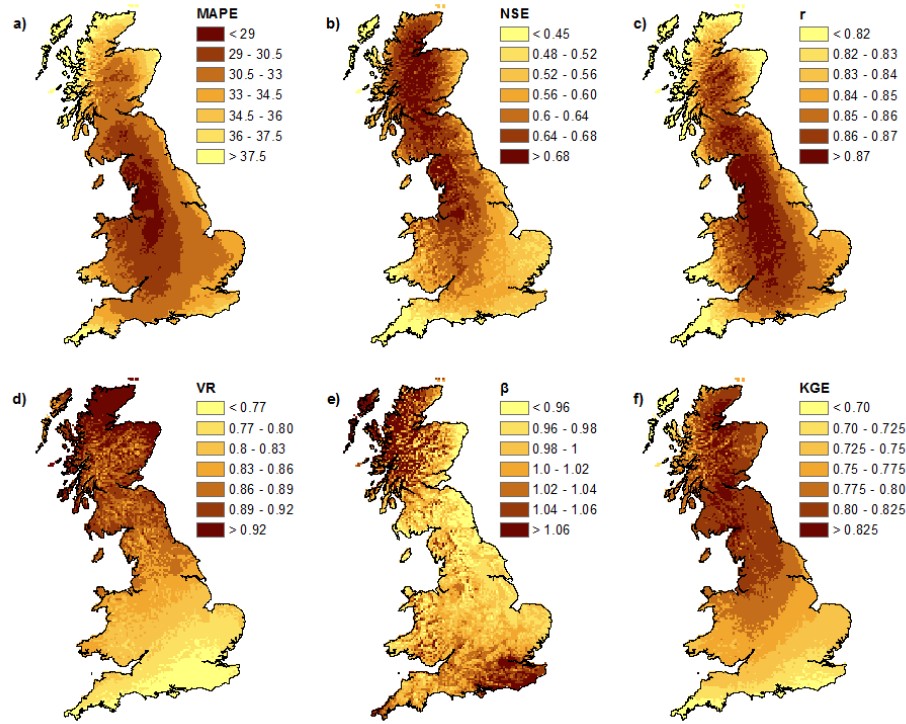

Figure 5: Grids of evaluation metrics for the new daily gridded PET dataset. The darker the colour, the better the performance for all metrics represented, except for the Bias ratio (β) (Fig.4e) where the middle-range colour is optimal.



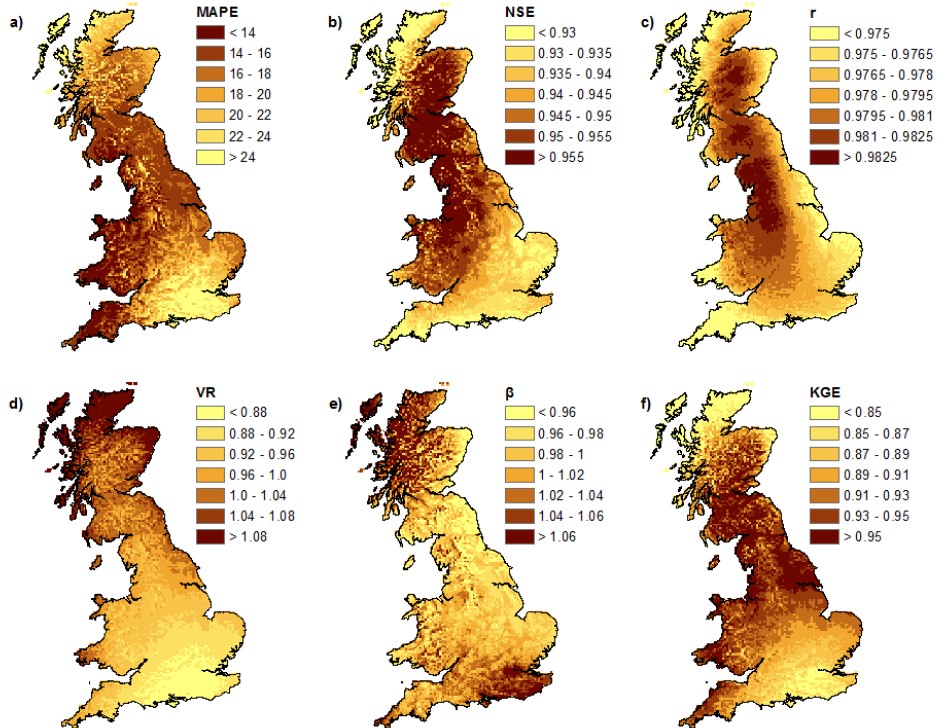

**Figure 6: Grids of evaluation metrics for the new monthly gridded PET dataset. The darker the colour, the better the performance for all metrics represented, except for the Variability Ratio (VR) (Fig.5d) and Bias ratio (β) (Fig.5e) where the middle-range colour is optimal.**

