# Peer review of "Historical gridded reconstruction of potential evapotranspiration for the UK"

_Earth System Science Data, 2017_

## Referee Comment (RC1) · Anonymous Referee #1 · 21 Feb 2018

Historical gridded reconstruction of potential evapotranspiration for the UK Tanguy et al.

This paper produces a gridded reconstruction (at daily and monthly timesteps) of PET for the UK (excluding Northern Ireland) for the period 1891-2015. The paper presents the selection of methods and decisions in producing the final dataset and assesses the performance of the selected approaches relative to a naive climatology and CHESS-PET as a surrogate for observations. The work produces a valuable dataset of practical utility, especially for river flow reconstructions. Overall the authors do a good job and I support publication. However I have a number of specific points that I wish the authors to consider before publication. These relate to both the underpinning science and uncertainties but also the presentation of the paper to help readers interpret what was

done in a clearer way.

Specific Comment

The work uses UKCP09 monthly temperature (1910-2015), together with a gridded dataset of monthly temperature from the historical drought project. I am left wondering about some details of the underlying temperature datasets – i) is there a decrease in station density underpinning these products the further back in time one goes? ii) does this affect the spatial distribution or errors in your dataset? iii) does the joining of both datasets create a break in the data, has this been checked? iv) has the underlying temperature data been homogenised? If so, how, if not what might this mean for the derived product here. So far as I can see these issues are not outlined or discussed in the paper. A fuller discussion on the uncertainties associated with the derived product is required in the discussion. You indicate that MAPE can be used to estimate uncertainties but only based on the selected method. There are other, potentially greater uncertainties that should be transparently laid out.

While the authors do outline the different temperature based PET methods in a table it would be beneficial to include a discussion of the main differences between each method in the text. Are these an exhaustive selection, if not, why these methods, why not others.

There are a lot of datasets/methods/calibration designs/verification methods used in the paper and at times it is hard to follow. Some effort at making the presentation clearer though signposting is necessary. The authors do include a work flow diagram but this too is complex. Perhaps a table describing the different datasets developed and why used would be useful. Perhaps the flow diagram could be split in two – separate for validation with some further detail to help interpretation in both parts.

I do not know what the pchip method is or how it performed, or why is was used above other approaches for disaggregation. Would use of another methods affect results? Pchip is mentioned in the abstract and once or twice in the paper but we have not

details about its application.

Minor Points/Technical corrections

Is there any influence of catchment size on the results?

In terms of extending to Northern Ireland, could reanalysis data be used here in future work. I think there is an onus on the authors to discuss how currently limitations may be overcome in future work. Abstract – needs to be reworked. You state that PET is needed at daily or shorter time step – do you mean longer? You examine monthly and daily, not hourly. I suggesting removing the word 'reconstructing' from line 3 of abstract – application of models before 1960 could be for a number of purposes – much flow data commences before PET data. Sentence commencing line 15 is too long, needs to be broken into at least two sentences. You need to tell the reader in the abstract what naive methods are. Line 25 -27 is perhaps too detailed for an abstract. What is pchip? Abstract also needs a final statement on envisaged uses of the dataset.

Introduction Is there a word missing from end of the first sentence?

Line 7 – suggest the word approaches rather than formulations. Also what are combination methods – used without any explanation in the first instance.

Line 13 – temperature as 'a' proxy

Page 3, line 1 – do you have a reference to support the claim that PET is mainly used for hydro modelling. If not suggest widely used rather than mainly.

Line 9 – as 'an' alternative

Line 12 – longest period in the UK – give us some context about the historical length of observations. Also maybe suggest that temp and precip are 'among' the those with longest records. Other variables such as Sea Level Pressure also have been rescued historically.

Actually, thinking about this, could reanalysis products that assimilate SLP observations be used to supplement this work and temporally extend the record in future work? Perhaps you could come to this potential (or not) in the discussion.

Line 13 – where only temperature 'data' are available.

Line 24 – for the international reader use of Great Britain, UK, NI can be confusing. Keep the terminology the same – suggest UK (incl. NI).

When introducing temperature data please include in the bracket the term henceforth. . ..eg. (henceforth CHESS-temp daily)

Page 4 line 8 – Because of the coarser temporal and spatial resolution of temperature data prior to 1961 – please give us some details on this and the PET dataset is dependent on this data.

Line 10-25 – you need to help the reader more in introducing these datasets and study design. The text is a little terse here and too brief. Some futher reasoning and justification required. Eg. above, line 23 (iii) – I am not sure exactly what is going on here.

Line 29 – are 7 daily datasets derived?

Page 5 – Clearer signposting needed in introducing methods. Please link to section in which the detail can be found.

Line 13 – am not sure about the use of quality assessment tests here – to me these mean homogeneity tests which is not the case in this paper. I think you are assessing performance?

Lines 17-20 – you can delete the first two sentences – repetition

Rather than commencing with 'Four main temperature-based equations were evaluated. . .' Start with seven and then differentiate.

Line 23 – what do you mean by a calibration procedure?

Line 28 – min max temps not always available historically either.

Page 6 – line 2 – what do you mean by time efficient?

Line 4 – can you provide some indication of range of catchments – size etc.

Line 13 – sentence beginning this line is long and confusing – break it into two.

Line 20 – delete of similar length

Line 21- were the assumptions of OLS checked?

Line 28 – can you call this forcing data – suggest temperature data

Page 7 line 7 – this sentence needs reworking- what about hydrological models? Is NSE a concept? Why might NSE be suited to assessing PET?

Page 8 – why these assessment criteria – later it becomes clear but state here.

Line 24 – 1P-GB introduced first time – at least link to the figure.

Page 9 – line 5-8 sentence too long, too many commas.

Line 12 delete in conclusion – this is not the conclusion

Am left wondering if reduction in temperature station density back in time is evident and if this affects results.

Page 10 – line 16 no need to present this correlation coefficient

Line 31 – give us the values of the more moderate performance

Page 11 – discussion and limitations needs to include fuller assessment of uncertainties . Line – 12 – please state what PM is here

Line 18 – replace would most likely be with are

Line 19 Great Britain? This include NI or Scotland?

Line 19 – when and where such high resolution. . ..

Line 23 – replace unique with calibrated

Line 26 – e.g. provide guidance

Page 12 – line 11 – move this finding (vi) to higher prominence.

Line 14 – replace metrics grid with gridded metrics

Line 19 The name of the dataset is a little misleading potentially – as it is stated it reads that it is calibrated for the UK over the period 1890-2015. Consider rewording.

I have not checked the references

Captions Fig 1 – caption needs to be more informative and help the reader interpret this complex figure

Fig 2 – same, could be more informative

Fig 3 – same point

Fig 5 – in caption 4e should be 5e. What is upper VR range, please relate to part of the text where indices are described.

Fig 6 – 5d should be 6d and same for 5e

―――――――――――

---

## Author Comment (AC1) · 23 Mar 2018

The authors' first response to reviewers: Anonymous Referee #1

The reviewer's comments are in black, and our response is in blue.

We would like to thank Referee #1 for his/her valuable and constructive comments which will help improve the manuscript. Below is our point-to-point response (in blue).

Historical gridded reconstruction of potential evapotranspiration for the UK Tanguy et al.

This paper produces a gridded reconstruction (at daily and monthly timesteps) of PET for the UK (excluding Northern Ireland) for the period 1891-2015. The paper presents the selection of methods and decisions in producing the final dataset and assesses the performance of the selected approaches relative to a naive climatology and CHESSPET as a surrogate for observations. The work produces a valuable dataset of practical utility, especially for river flow reconstructions. Overall the authors do a good job and I support publication. However I have a number of specific points that I wish the authors to consider before publication. These relate to both the underpinning science and uncertainties but also the presentation of the paper to help readers interpret what was done in a clearer way.

We would like to clarify that the new PET dataset **does** cover Northern Ireland (NI). It is the high resolution CHESS PET dataset (available for 1961-2015) that is not available for NI. As CHESS PET was used for the assessment, the performance metrics could not be calculated for NI, but the data was produced, and we expect performance to be similar due to geographic proximity. We will revise the text to make sure this is clear.

**Specific Comment**

The work uses UKCP09 monthly temperature (1910-2015), together with a gridded dataset of monthly temperature from the historical drought project. I am left wondering about some details of the underlying temperature datasets – i) is there a decrease in station density underpinning these products the further back in time one goes? ii) does this affect the spatial distribution or errors in your dataset? iii) does the joining of both datasets create a break in the data, has this been checked? iv) has the underlying temperature data been homogenised? If so, how, if not what might this mean for the derived product here. So far as I can see these issues are not outlined or discussed in the discussion. You indicate that MAPE can be used to estimate uncertainties but only based on the selected method. There are other, potentially greater uncertainties that should be transparently laid out.

As pointed out by Referee #1, we have only investigated the uncertainties coming from the method used to calculate PET, but not from the underlying temperature dataset used. We will add a paragraph in the manuscript to discuss the use of the underlying temperature data, and its implication on the final uncertainty. We will also more fully discuss all these issues in the supplementary information.

More specifically:

- There is indeed a change in station density underpinning the temperature grids. According to information provided by the Met Office, the station density gradually increased from 74 stations across the country in 1891 to a peak of 672 in the mid-1990s, after which it decreased again to reach a total of 355 stations in 2015.
- Legg (2015) has investigated extensively the effect of network density on the error in gridded dataset in the UK, and his results suggest that the change in density observed here would only lead to a minor increase in error in temperature. An increase in the root-mean-square error of less than 0.2°C is observed for most cases when the network density changes from 570 to 75 stations across the UK. This reflects the spatial coherence in the temperature data. We have conducted a sensitivity analysis of PET on errors in input temperature, when using McGuinness-Bordne equation. We have found that a +/- 0.2°C in input temperature translates into a 0.5% to 2% difference (with an average of 0.8%) in PET estimation. We consider these differences negligible in comparison to the uncertainties arising from the PET method itself.
- iii) The joining of both datasets does not create a break in the dataset, as the exact same methodology was applied to derive the grids (method described in Perry and Hollis, 2005). The dataset previously only existed from 1910, as the density of stations was too sparse prior to that. However, within Historic Droughts project, historic data has been rescued and digitised by the Met Office, which has allowed them to extend the gridded temperature back to 1891.
- iv) The gridded temperature product used in this study is a standard national product produced by the UK Met Office National Climate Information Centre (NCIC). All the underlying stations are part of the climatological network administered by the Met Office, which are subject to common observation methods, regular site inspections and instrument calibration. The data is also subject to quality control procedures prior to archiving. For all this, Prior and Perry (2014) concluded that, even if individual station records were not tested for homogeneity, the effect of inhomogeneities in the gridded product is considered minimal. The interpolation and regression methods used to create the gridded product, which takes into account factors such as latitude and longitude, altitude and terrain shape, coastal influence, and urban land use, reduces the impact of station openings and closures on homogeneity, although it can't be removed entirely, especially in areas of complex topography or sparse station coverage (Perry and Hollis, 2005).

**References:**

Legg, T. (2015), Uncertainties in gridded area-average monthly temperature, precipitation and sunshine for the United Kingdom. Int. J. Climatol., 35: 1367–1378. doi:10.1002/joc.4062

Prior, M. J. and Perry, M. C. (2014), Analyses of trends in air temperature in the United Kingdom using gridded data series from 1910 to 2011. Int. J. Climatol., 34: 3766-3779. doi:10.1002/joc.3944

While the authors do outline the different temperature based PET methods in a table it would be beneficial to include a discussion of the main differences between each method in the text. Are these an exhaustive selection, if not, why these methods, why not others.

We will add some text to highlight the main differences between each method in the manuscript. The physical basis for estimating evaporation using temperature alone is that both terms of the combination equation (the energy required to sustain evaporation and the energy removed from the surface as water vapour) are generally related to temperature (Shuttleworth, 1993). The main difference between the different temperature-based formulations, lies in the way temperature is linked to PET to simulate the effect of the full set of variables normally required in the combination equations. Most temperature-based equations use day length or related variables (Hamon, 1961; Blaney and Criddle, 1950; Kharrufa, 1985; MOHYSE: Fortin, 2006 and Thornwaite, 1948), except McGuinness and Bordne (1972), and the derived Oudin (2005)'s equation which use extraterrestrial radiation instead. Blaney-Criddle equation has also an additional parameter k, which depends on crop type. Most of these equations were developed for the USA, except MOHYSE (which was developed in Quebec), Kharrufa (developed for arid regions) and Oudin (developed in Australia, USA and France).

We have tested all widely used temperature-based PET equations that we were aware of, using mean temperature data as the only climatic variable.

There are a lot of datasets/methods/calibration designs/verification methods used in the paper and at times it is hard to follow. Some effort at making the presentation clearer through signposting is necessary. The authors do include a work flow diagram but this too is complex. Perhaps a table describing the different datasets developed and why used would be useful. Perhaps the flow diagram could be split in two – separate for validation with some further detail to help interpretation in both parts.

We realise the paper can be difficult to follow due to multiple datasets/methods/assessments. We will make sure there will be more signposting in the revised manuscript to help the readers' comprehension. We will add the following table in the supplementary information (new table A3) to summarise datasets used, and we will replace the current flow diagram (Figure 1) with the following revised version (new Figure 1). The text in the manuscript will also be modified to better follow the Stages described in the new figure.

Table A3: Summary of temperature and PET datasets used in this study a) temperature data to investigate effect of temporal distribution of data on the output PET estimation, b) temperature data to investigate effect of spatial resolution of the data on the output PET estimation, and c) PET data used to calibrate and evaluate equation and final PET output.

a) Temperature datasets used as input data for temperature-based PET equations. Multiple versions were used to investigate the effect of temporal distribution of the data on output PET estimation

| Dataset short name   | Resolution    | Description                                 | Comment                  |  |  |
|----------------------|---------------|---------------------------------------------|--------------------------|--|--|
| CHESS-temp daily     | 1 km x 1 km   | CHESS-met high resolution mean daily        | "Best" available         |  |  |
|                      | Daily         | temperature: Part of a larger dataset       | gridded daily            |  |  |
|                      |               | developed by CEH for environment            | temperature data for     |  |  |
|                      |               | modelling applications, available for 1961- | Great Britain            |  |  |
|                      |               | 2015                                        |                          |  |  |
| CHESS-temp clim      | 1 km x 1 km   | CHESS daily mean temperature                | Default option that      |  |  |
|                      | Daily         | climatology: Long term average (1961-       | could be used even if no |  |  |
|                      |               | 1990) of daily mean temperature, derived    | temperature data were    |  |  |
|                      |               | from CHESS-temp daily                       | available                |  |  |
| CHESS-temp monthly I | 1 km x 1 km   | CHESS daily mean temperature derived        |                          |  |  |
|                      | Monthly       | from monthly averages, constant during      |                          |  |  |
|                      | disaggregated | the month. Step changes in temperature      |                          |  |  |
|                      | to daily      | between consecutive months                  | To investigate whether   |  |  |
| CHESS-temp monthly   | 1 km x 1 km   | CHESS daily mean temperature derived        | temporal disaggregation  |  |  |
| II                   | Monthly       | from monthly averages, interpolated using   | method (from monthly     |  |  |
|                      | disaggregated | pchip                                       | to daily) has an effect  |  |  |
|                      | to daily      |                                             | on output PET            |  |  |
| CHESS-temp monthly   | 1 km x 1 km   | CHESS daily mean temperature derived        | estimation               |  |  |
| III                  | Monthly       | from monthly averages, disaggregated to     |                          |  |  |
|                      | disaggregated | daily using CHESS daily mean                |                          |  |  |
|                      | to daily      | temperature climatology pattern             |                          |  |  |

b) Temperature datasets used to assess spatial resolution for the best performing PET method

| Dataset short name        | Resolution                                          | Description                                                                                  | Comment      |  |
|---------------------------|-----------------------------------------------------|----------------------------------------------------------------------------------------------|--------------|--|
| UKCP09-temp monthly
I  | 5 km x 5 km
Monthly
disaggregated
to daily | UKCP09 daily mean temperature derived
from monthly averages, constant during
the month | Two temporal |  |
| UKCP09-temp monthly
II | 5 km x 5 km
Monthly
disaggregated
to daily | UKCP09 daily mean temperature derived
from monthly averages, interpolated using
pchip  | tested.      |  |

c) PET datasets used to calibrate the equations and assess the output PET

| Dataset short name | Resolution  | Description                    | Use                                |
|--------------------|-------------|--------------------------------|------------------------------------|
| CHESS-PM           | 1 km x 1 km | CHESS-PET 1-km grids, daily    | 1) calibration of the temperature- |
|                    | Daily and   | (and monthly) time series      | based PET equations (1961-1990)    |
|                    | monthly     | available for 1961-2015,       | 2) Evaluation of the equations     |
|                    |             | calculated using the Penman-   | (1991-2012)                        |
|                    |             | Monteith (PM) equation for     | 3) Evaluation of the final gridded |
|                    |             | FAO-defined well-watered grass | product (1991-2012)                |
| CHESS-PM           | 1 km x 1 km | Daily (and monthly) PET long   | used as a 'naïve method' against   |
| climatology        | Daily and   | term average, calculated from  | which the PET reconstruction       |
|                    | monthly     | CHESS-PM for 1961 to 1990      | methodology can be tested to       |
|                    |             |                                | assess performance                 |

---

## Referee Comment (RC2) · M. McCarthy (Referee) · 4 Apr 2018

This paper documents the development of a unique and valuable PET dataset for application in historical hydrological reconstructions for the UK. For full disclosure I have been involved as a subcontractor within the Historic Droughts project that has supported this work. My role was providing extended monthly temperature datasets with newly digitised climate data that are used and referenced in this paper. However I did not have any direct involvement in the specific work in question, the development of the PET dataset that is being described, or the writing of the paper.

In my opinion this paper provides a very clear description of the production process of this dataset, the calibration and evaluation framework that informed and justify the

methodological decisions that were made. This is important because there are significant assumptions required for such a temperature only derivation of PET. Therefore I recommend this work is accepted for publication.

My only significant comments would be: 1) While the paper provides sufficient comparison of different potential approaches, in section 5 the paper does not provide much context for how the uncertainties and limitations of this PET dataset might impact or be handled by subsequent application in hydrological modelling of 19th and early 20th century. Are there any firmer recommendations or quantifications the authors can make in that regard?

2) Will a significant factor in the evaluation metrics be the seasonal cycle? Would this explain why some of the differences between the performance of the choice of temperature data is marginal, because they all have good representation of the seasonality and the daily variance is of secondary importance?

3) I did find I had to keep referring back to section 2 to remind myself which specific data were being discussed. A summary table of datasets would help rather than just a list I think.

4) In section 2 it is probably worth being more explicit about what temperature data. For UKCP09 and HistDrought the monthly mean temperature is derived from the average of daily Tmax and Tmin averaged across the month at each contributing station and then stations with no more than 2 missing days within a calendar month are gridded as per Perry and Hollis (2005).

Minor points:

1) Page 4, line 5: could include reference to Legg (2014) https://rmets.onlinelibrary.wiley.com/doi/abs/10.1002/joc.4062 that documents the spatial sampling uncertainty in the monthly gridded data.

2) Page 4, lines 10-30. I think this could be reworded slightly with a table to lay out the

datasets in a slightly clearer way.

3) Page 4, line 18. Provide either a description or reference to pchip.

4) Page 5, line 27. It is not clear why this max/min constraint is important in this context. This dataset only covers the observational period 1891-2015. What forecasts are used?

5) Page 6, line 14-16. Suggest including shorthand used elsewhere. e.g. "from a global parameterisation (GB) leading to a single equation (1P) for all 43 catchments (1P-GB)"

6) Page 7, line 4: How often is PET 0. Does this skew the MAPE score in certain situations?

7) Page 8, line 13: Figure 4 includes "no calibration" how does this differ from "uncalibrated" I didn't quite follow this.

8) Page 8, line 25: I don't think 'forcing' is the right term here. perhaps just data?

9) Page 8, line 26: Referring to my comment above, is the small day-to-day variability in relation to the magnitude of the seasonal cycle and therefore why the differences are only marginal? Does this have implications for any particular use-cases?

10) Page 9, line 11: I'm afraid I lost the thread of this a little. Intuitively I agree it seems surprising that this is the case (how significant is the difference?), but not sure specifically what PET estimate is closer to what observed data in the final sentence?

---

## Author Comment (AC2) · 20 Apr 2018

**The authors' first response to reviewers: Mark McCarthy, Referee #2**

The reviewer's comments are in black, and our response is in blue.

This paper documents the development of a unique and valuable PET dataset for application in historical hydrological reconstructions for the UK. For full disclosure I have been involved as a subcontractor within the Historic Droughts project that has supported this work. My role was providing extended monthly temperature datasets with newly digitised climate data that are used and referenced in this paper. However I did not have any direct involvement in the specific work in question, the development of the PET dataset that is being described, or the writing of the paper.

In my opinion this paper provides a very clear description of the production process of this dataset, the calibration and evaluation framework that informed and justify the methodological decisions that were made. This is important because there are significant assumptions required for such a temperature only derivation of PET. Therefore I recommend this work is accepted for publication.

We would like to thank Mark McCarthy for reviewing this paper and for his positive and constructive comments, which will contribute to improve the manuscript.

My only significant comments would be: 1) While the paper provides sufficient comparison of different potential approaches, in section 5 the paper does not provide much context for how the uncertainties and limitations of this PET dataset might impact or be handled by subsequent application in hydrological modelling of 19th and early 20$^{th}$ century. Are there any firmer recommendations or quantifications the authors can make in that regard?

Regarding the increased uncertainty due to lower station density in the earlier period (late 19$^{th}$ and early 20$^{th}$ century), some additional detail will be added to the manuscript:

"According to information provided by the Met Office, the station density gradually increased from 74 stations across the country in 1891 to a peak of 672 in the mid-1990s, after which it decreased again to reach a total of 355 stations in 2015. Legg (2015) has investigated extensively the effect of network density on the error in gridded dataset in the UK, and his results suggest that the change in density observed here would only lead to a minor increase in error in temperature. An increase in the root-mean-square error of less than 0.2°C is observed for most cases when the network density changes from 570 to 75 stations across the UK. This reflects the spatial coherence in the temperature data.

A sensitivity analysis of McGuinness-Bordne PET on errors in input temperature was conducted. It was found that a +/- 0.2°C in input temperature translates into a 0.5% to 2% difference (with an average of 0.8%) in PET estimation. We consider these differences negligible in comparison to the uncertainties arising from the PET method itself."

We agree that offering more guidance on how to use the data would be beneficial. We will add references to provide some context to users about how the uncertainties of the PET dataset might

impact hydrological and other applications. Section 5 will be modified to include the following information:

"While uncertainties in the PET dataset are quite large, especially in the daily version, the impact it might have will depend on the intended purpose of the data.

For hydrological applications, the choice of PET equation was shown to affect the estimated streamflow when using hydrological models (Seiller and Anctil, 2016), in particular at high and low flows (Zahra Samadi, 2016). However, several studies show that hydrological models are much more sensitive to errors in rainfall than to errors in PET, especially in temperate climate such as the UK (Paturel et al., 1995, Guo et al., 2017, Bastola et al., 2011). Furthermore, other studies (Bai et al., 2015, Seiller and Anctil, 2016) show that hydrological model parameter calibration can eliminate the influences of different PET inputs on runoff simulations. Therefore, the historic PET dataset is considered particularly suitable for use in hydrological models, especially if these are being calibrated using this dataset, as the impact of PET uncertainties will be small compared to those of rainfall. It's also worth mentioning that the McGuinness-Bordne equation used to derive the historic PET dataset was calibrated against CHESS-PM. There is no systematic bias (bias ratio ≈ 1, see Fig.5 and 6) between the two datasets. The use of the historic PET data would therefore be adequate in hydrological models that have been calibrated using CHESS-PM, but re-calibration would be recommended if any other PET source was used in the original calibration.

For crop modelling, greater caution is required as modelled crop yield is highly sensitive to the choice of PET model (Balkovic et al., 2013, Liu et al., 2016, Luo et al., 2009).

For macroecology and biogeography studies, Fisher et al. (2011) have produced a global 'guide to choosing an ET model for geographical ecology', according to the climate zone of the study area. For temperate climate such as the UK, their conclusion is that any PET model type (temperature-based, radiation-based or combination) is equally adequate for its use in biodiversity modelling. Therefore, the historic PET dataset would be appropriate for this type of application.

Regarding the derivation of drought indices which use PET, some seem insensitive to the choice of PET model, such as the Reconnaissance Drought Index (RDI, Tsakiris et al., 2007) as demonstrated by Vangelis et al. (2013); whereas for others such as the standardized Precipitation-Evapotranspiration Index (SPEI, Vicente-Serrano, 2010) or the Palmer Drought Severity Index (PDSI, Palmer, 1965), different formulations of PET have a significant impact on the result (for SPEI: Begueria et al., 2013, Stagge et al., 2014; for PDSI: Sheffield et al., 2012), although less importantly in humid areas such as the UK (Begueria et al., 2013). Therefore, the impact of uncertainties in PET for deriving drought indices will depend on the choice of index.

In general, for the use of the historic PET dataset to derive drought indices, or any other application not mentioned above, we would recommend to compare results over the more recent period (1961-2015) using (i) CHESS-PM and (ii) the historic temperature-based PET to estimate the impact of uncertainties in PET on results. This way, the users can truly assess the sensibility of their specific application to the errors in PET, investigate how the uncertainties propagate in their model, and make an informed decision on whether the historic PET dataset is suitable for their needs or not."

**References:**

Legg, T. (2015), Uncertainties in gridded area-average monthly temperature, precipitation and sunshine for the United Kingdom. Int. J. Climatol., 35: 1367–1378. doi:10.1002/joc.4062

G. Seiller & F. Anctil (2016) How do potential evapotranspiration formulas influence hydrological projections?, Hydrological Sciences Journal, 61:12, 2249-2266, DOI: 10.1080/02626667.2015.1100302

S. Zahra Samadi, Assessing the sensitivity of SWAT physical parameters to potential evapotranspiration estimation methods over a coastal plain watershed in the southeastern United States, Hydrology Research Jul 2016, nh2016034; DOI: 10.2166/nh.2016.034

J.E. Paturel, E. Servat, A. Vassiliadis, Sensitivity of conceptual rainfall-runoff algorithms to errors in input data — case of the GR2M model, Journal of Hydrology, Volume 168, Issues 1–4, 1995, Pages 111-125, ISSN 0022-1694, https://doi.org/10.1016/0022-1694(94)02654-T.

Danlu Guo, Seth Westra, Holger R. Maier, Use of a scenario-neutral approach to identify the key hydro-meteorological attributes that impact runoff from a natural catchment, Journal of Hydrology, Volume 554, 2017, Pages 317-330, ISSN 0022-1694, https://doi.org/10.1016/j.jhydrol.2017.09.021.

Satish Bastola, Conor Murphy, John Sweeney, The sensitivity of fluvial flood risk in Irish catchments to the range of IPCC AR4 climate change scenarios, Science of The Total Environment, Volume 409, Issue 24, 2011, Pages 5403-5415, ISSN 0048-9697, https://doi.org/10.1016/j.scitotenv.2011.08.042.

Bai, P., X. Liu, T. Yang, F. Li, K. Liang, S. Hu, and C. Liu, 2016: Assessment of the Influences of Different Potential Evapotranspiration Inputs on the Performance of Monthly Hydrological Models under Different Climatic Conditions. J. Hydrometeor., 17, 2259–2274, https://doi.org/10.1175/JHM-D-15-0202.1

Juraj Balkovič, Marijn van der Velde, Erwin Schmid, Rastislav Skalský, Nikolay Khabarov, Michael Obersteiner, Bernhard Stürmer, Wei Xiong, Pan-European crop modelling with EPIC: Implementation, up-scaling and regional crop yield validation, Agricultural Systems, Volume 120, 2013, Pages 61-75, ISSN 0308-521X, https://doi.org/10.1016/j.agsy.2013.05.008.

Wenfeng Liu, Hong Yang, Christian Folberth, Xiuying Wang, Qunying Luo, Rainer Schulin, Global investigation of impacts of PET methods on simulating crop-water relations for maize, Agricultural and Forest Meteorology, Volume 221, 2016, Pages 164-175, ISSN 0168-1923, https://doi.org/10.1016/j.agrformet.2016.02.017.

Luo, W., Jing, W., Jia, Z. et al. Sci. China Ser. E-Technol. Sci. (2009) 52: 3315. https://doi.org/10.1007/s11431-009-0349-0

Fisher, J. B., Whittaker, R. J. and Malhi, Y. (2011), ET come home: potential evapotranspiration in geographical ecology. Global Ecology and Biogeography, 20: 1-18. doi:10.1111/j.1466-8238.2010.00578.x

Tsakiris, G., Pangalou, D. & Vangelis, H. Water Resour Manage (2007) 21: 821. https://doi.org/10.1007/s11269-006-9105-4

H. Vangelis, D. Tigkas, G. Tsakiris, The effect of PET method on Reconnaissance Drought Index (RDI) calculation, Journal of Arid Environments, Volume 88, 2013, Pages 130-140, ISSN 0140-1963, https://doi.org/10.1016/j.jaridenv.2012.07.020.

Vicente-Serrano, S.M., S. Beguería, and J.I. López-Moreno, 2010: A Multiscalar Drought Index Sensitive to Global Warming: The Standardized Precipitation Evapotranspiration Index. J. Climate, 23, 1696–1718, https://doi.org/10.1175/2009JCLI2909.1

Palmer WC. 1965. Meteorological droughts. U.S. Department of Commerce Weather Bureau Research Paper 45, 58.

Stagge, J., Tallaksen, L., Xu, C.-Y., Van Lanen, H. (2014). Standardized precipitation-evapotranspiration index (SPEI): Sensitivity to potential evapotranspiration model and parameters. Hydrology in a changing world: Environmental and Human Dimensions. Proceedings of FRIEND-Water 2014, Montpellier, France, October 2014 (IAHS Publ. 363, 2014).

Beguería, S., Vicente-Serrano, S. M., Reig, F. and Latorre, B. (2014), Standardized precipitation evapotranspiration index (SPEI) revisited: parameter fitting, evapotranspiration models, tools, datasets and drought monitoring. Int. J. Climatol., 34: 3001-3023. doi:10.1002/joc.3887

2) Will a significant factor in the evaluation metrics be the seasonal cycle? Would this explain why some of the differences between the performance of the choice of temperature data is marginal, because they all have good representation of the seasonality and the daily variance is of secondary importance?

Yes, this is partly what we were trying to explain in line 25-30, page 8:

"Daily temperature forcing only performs marginally better than forcing based on monthly temperature time series. This might be explained by the small day-to-day variability in temperature fields (and hence, in any resulting PET field) compared with other climate variables such as wind speed, humidity or radiation, which provide a much larger contribution to the daily variability of PET than temperature. The effect of artificial daily pattern introduced by temporal disaggregation of monthly temperature is in fact small compared with the error introduced by using temperature-only forcing to estimate PET. This is illustrated in Figure A1 (supplementary material)."

But we will add some further comment to make this point clearer:

"The temperature seasonal variability is a main component to the PET, and is well captured by monthly values, with sub-monthly values only adding some noise. This is why the choice of temperature data has only a marginal effect, because the daily variance is of secondary importance in comparison to an accurate representation of the seasonality."

3) I did find I had to keep referring back to section 2 to remind myself which specific data were being discussed. A summary table of datasets would help rather than just a list I think.

We acknowledge that the numerous datasets used can lead to confusion. Therefore, we will add to the supplementary information the following summary table (Table A3) of all datasets used.

**Table A3: Summary of temperature and PET datasets used in this study a) temperature data to investigate effect of temporal distribution of data on the output PET estimation, b) temperature data to investigate effect of spatial resolution of the data on the output PET estimation, and c) PET data used to calibrate and evaluate equation and final PET output.**

*a) Temperature datasets used as input data for temperature-based PET equations. Multiple versions were used to investigate the effect of temporal distribution of the data on output PET estimation*

| Dataset short name | Resolution | Description | Comment |
|---|---|---|---|
| CHESS-temp daily | 1 km x 1 km Daily | CHESS-met high resolution mean daily temperature: Part of a larger dataset developed by CEH for environment modelling applications, available for 1961-2015 | "Best" available gridded daily temperature data for Great Britain |
| CHESS-temp clim | 1 km x 1 km Daily | CHESS daily mean temperature climatology: Long term average (1961-1990) of daily mean temperature, derived from CHESS-temp daily | Default option that could be used even if no temperature data were available |
| CHESS-temp monthly I | 1 km x 1 km Monthly disaggregated to daily | CHESS daily mean temperature derived from monthly averages, constant during the month. Step changes in temperature between consecutive months | To investigate whether temporal disaggregation method (from monthly to daily) has an effect on output PET estimation |
| CHESS-temp monthly II | 1 km x 1 km Monthly disaggregated to daily | CHESS daily mean temperature derived from monthly averages, interpolated using pchip | |
| CHESS-temp monthly III | 1 km x 1 km Monthly disaggregated to daily | CHESS daily mean temperature derived from monthly averages, disaggregated to daily using CHESS daily mean temperature climatology pattern | |

*b) Temperature datasets used to assess spatial resolution for the best performing PET method*

| Dataset short name | Resolution | Description | Comment |
|---|---|---|---|
| UKCP09-temp monthly I | 5 km x 5 km Monthly disaggregated to daily | UKCP09 daily mean temperature derived from monthly averages, constant during the month | Two temporal disaggregation methods tested. |
| UKCP09-temp monthly II | 5 km x 5 km Monthly disaggregated to daily | UKCP09 daily mean temperature derived from monthly averages, interpolated using pchip | |

*c) PET datasets used to calibrate the equations and assess the output PET*

| Dataset short name | Resolution | Description | Use |
|---|---|---|---|
| CHESS-PM | 1 km x 1 km Daily and monthly | CHESS-PET 1-km grids, daily (and monthly) time series available for 1961-2015, calculated using the Penman-Monteith (PM) equation for FAO-defined well-watered grass | 1) calibration of the temperature-based PET equations (1961-1990) 2) Evaluation of the equations (1991-2012) 3) Evaluation of the final gridded product (1991-2012) |
| CHESS-PM climatology | 1 km x 1 km Daily and monthly | Daily (and monthly) PET long term average, calculated from CHESS-PM for 1961 to 1990 | used as a 'naïve method' against which the PET reconstruction methodology can be tested to assess performance |

4) In section 2 it is probably worth being more explicit about what temperature data. For UKCP09 and HistDrought the monthly mean temperature is derived from the average of daily Tmax and Tmin averaged across the month at each contributing station and then stations with no more than 2 missing days within a calendar month are gridded as per Perry and Hollis (2005).

This information will be added to section 2.

Minor points:

1) Page 4, line 5: could include reference to Legg (2014) https://rmets.onlinelibrary.wiley.com/doi/abs/10.1002/joc.4062 that documents the spatial sampling uncertainty in the monthly gridded data.

This reference will be added to the revised manuscript, together with a short discussion from Legg (2014)'s results on the effect of change in network density on the output gridded product (as described in response to first comment).

2) Page 4, lines 10-30. I think this could be reworded slightly with a table to lay out the datasets in a slightly clearer way.

Table A3 shown earlier will be added to the supplementary information.

Additionally, page 4, lines 10-27 will be replaced by the following text:

'Prior to 1961, temperature data is only available at a 5km spatial resolution and monthly time-step. Because of this coarser temporal and spatial resolution of temperature data in the earlier period, alternative datasets were generated and used in the analysis to quantify the sensitivity of PET derivation to temperature input, and are summarised in table A3 (a and b) in the supplementary information:

- CHESS daily mean temperature climatology (1-km grids) (CHESS-temp clim): long term average (1961-1990) of daily mean temperature, derived from CHESS-temp daily. This provides a default option that could be used even if no temperature data were available in the past (or future). This gives a day-to-day variability pattern of temperature throughout the year, which is then repeated every year.
- CHESS daily mean temperature derived from monthly averages (1km grids). Different methods to disaggregate monthly temperature into daily data were tested:
  (i) Constant temperature during the month (CHESS-temp monthly I). This means there are step changes in temperature between consecutive months.
  (ii) Interpolated using pchip (piecewise cubic hermite interpolating polynomial) method for a smooth transition between months (CHESS-temp monthly II). Pchip stands for Piecewise Cubic Hermite Interpolating Polynomial, which is an interpolation method in which a cubic polynomial approximation is assumed over each subinterval. Arandiga et al. (2016) describe this interpolation scheme in detail together with its advantages, mainly that it is both accurate (preserves values at the nodes) and preserves monotonicity. Pchip was selected for the present study because (i) the fitted curve passes through observed values at inflexion points unlike spline or quadratic methods, for example, and (ii) it does not require re-fitting when the period of application is extended as each subinterval is treated separately.

(iii) Disaggregated to daily using CHESS daily mean temperature climatology pattern (CHESS-temp monthly III). The daily relative variation in temperature follows the climatology, but for each month, the daily values are adjusted so that monthly mean temperatures are correct. In other words, CHESS daily climatology data is shifted uniformly so the monthly mean temperature matches the CHESS monthly temperature data.

- UKCP09 daily mean temperature (5-km grids) derived from monthly averages. Two different methods to disaggregate monthly temperature into daily data were tested:
  (i) Constant during the month (UKCP09-temp monthly I).
  (ii) Interpolated using pchip method (UKCP09-temp monthly II).'

3) Page 4, line 18. Provide either a description or reference to pchip.

The following text will be added to the manuscript (as shown in response to previous point):

"Pchip stands for Piecewise Cubic Hermite Interpolating Polynomial, which is an interpolation method in which a cubic polynomial approximation is assumed over each subinterval. Arandiga et al. (2016) describe this interpolation scheme in detail together with its advantages, mainly that it is both accurate (preserves values at the nodes) and preserves monotonicity. Pchip was selected for the present study because (i) the fitted curve passes through observed values at inflexion points unlike spline or quadratic methods, for example, and (ii) it does not require re-fitting when the period of application is extended as each subinterval is treated separately."

Reference: F. Aràndiga, R. Donat, M. Santágueda, 2016, The PCHIP subdivision scheme, Applied Mathematics and Computation, Volume 272, Part 1, Pages 28-40, ISSN 0096-3003, https://doi.org/10.1016/j.amc.2015.07.071.

4) Page 5, line 27. It is not clear why this max/min constraint is important in this context. This dataset only covers the observational period 1891-2015. What forecasts are used?

We selected low-data demanding methods that could be easily reproduced and extended in cases of minimal data availability. Although the dataset described in this paper only covers the observational period, we also considered the applications of this method for forecasting. For the UK, the Met Office currently sends operationally average UK temperature forecasts (for 1-3months) for the production of the UK Hydrological Outlook (UKHO). Minimum and maximum temperatures are not included in the current seasonal forecasts and not used for the derivation of the UKHO, which is why we limited our study to temperature-based PET equations that uses mean temperature as input. Future work could complement the current study by including the evaluation of PET formulation using Tmin and Tmax.

5) Page 6, line 14-16. Suggest including shorthand used elsewhere. e.g. "from a global parameterisation (GB) leading to a single equation (1P) for all 43 catchments (1P-GB)"

Will be added.

6) Page 7, line 4: How often is PET 0. Does this skew the MAPE score in certain situations?

PET is equal to 0 about 3% of the time. The frequency is not high enough to skew the MAPE score.

7) Page 8, line 13: Figure 4 includes "no calibration" how does this differ from "uncalibrated" I didn't quite follow this.

'uncalibrated' models here refer to the three equations which were tested but not suitable for calibration (Oudin, MOHYSE and Thornwaite, Eq 5 to 7 in table 1).

We will rephrase page 8, line 13, to make this clearer:

'- models that were not calibrated in this study, i.e. Oudin, MOHYSE and Thornwaite (Eq 5 to 7 in table 1)'

8) Page 8, line 25: I don't think 'forcing' is the right term here. perhaps just data?

'forcing' will be replaced by 'data'

9) Page 8, line 26: Referring to my comment above, is the small day-to-day variability in relation to the magnitude of the seasonal cycle and therefore why the differences are only marginal? Does this have implications for any particular use-cases?

We have mostly covered this point in our responses to the two first comments.

In addition, we can say that:

The magnitude in the seasonal cycle has a greater impact than the small day-to-day variability in temperature, which explains why the differences in performance are only marginal. Regarding implications for particular use-cases, any application looking specifically at daily variability in PET should take into account that the PET dataset produced here is a smoothed version of reality. This is true for applications such as the estimation of daily water balance, flood peaks, crop water demand, among others.

In many applications, PET is used to estimate Actual Evapotranspiration (AET). AET is equal to PET only if there is no limitation in water (soil moisture) and there is enough energy to evaporate the water (radiation). PET thus represents the upper limit of AET. In radiation-limited or water-limited regions, AET is smaller than PET, hence the day-to-day variability of PET is less important.

Temperature-based PET equations per se already produce a much smoother version than 'real' PET time series (CHESS-PM, see Figure A1 in supplementary information). The added simplification coming from using monthly temperature, in which the daily variability of temperature is not captured, has only a minor additional effect on the overall performance.

10) Page 9, line 11: I'm afraid I lost the thread of this a little. Intuitively I agree it seems surprising that this is the case (how significant is the difference?), but not sure specifically what PET estimate is closer to what observed data in the final sentence?

We thank Mark for pointing out this issue. This has made us realised there is an error in the text: it is actually McGuinness-Bordne equation using CHESS-temp clim which produces a smoother time series than using CHESS-PM climatology (and not the other way around as it was in the original manuscript).

We will rephrase this sentence to correct the error and make the interpretation clearer, and we will also add the following figure (Fig.AX) to the supplementary information to illustrate what we are trying to say here.

The new text will be:

"A surprising result is that, in the absence of any climate data available, calibrating McGuinness-Bordne equation with CHESS-temp clim (long-term daily temperature climatology) outperforms using CHESS-PM climatology. NSE scores are equivalent for both approaches but MAPE is worse for the latter. The two approaches give similar results, but running McGuinness-Bordne equation using CHESS-temp clim produces smoother time series than using directly CHESS-PM climatology. The latter displays random noise which explains the larger values of MAPE compared to the smoother version. This is illustrated in Fig. AX of the supplementary information."

[Figure]

Fig.AX: Daily PET time series for an example catchment (23001), year 1991, to illustrate the differences between (i) CHESS-PM, proxy to observed PET (grey line), (ii) PET calculated using McGuinness-Bordne equation, using CHESS daily temperature climatology (long term average from 1961-1990) (blue line), and (iii) CHESS-PM daily climatology (long term average from 1961-1990) (red line).